# Influence of Nature Reserves on the Energy Consumption Structure of Local Farmers

**DOI:** 10.3390/ijerph191911955

**Published:** 2022-09-21

**Authors:** Ke Chen, Runze Li, Yang Wang

**Affiliations:** College of Economics and Management, Shenyang Agricultural University, Shenyang 110866, China

**Keywords:** energy consumption structure, environmental cognition of farmers, nature reserves, propensity score matching, surrounding farmers

## Abstract

The energy consumption of farmers residing around nature reserves is an important factor that affects the coordinated development of nature reserves and the surrounding communities. The optimization of the energy consumption structure of farmers around nature reserves is important for maintaining the resources and environment of nature reserves and saving natural resources. Based on the microscopic survey data for the energy consumption structure of 956 rural households around six nature reserves in Liaoning province, a multiple linear regression model was used in this study to match tendency scores and empirically examine the impacts of regulatory policies of nature reserves on the energy consumption of rural households in Liaoning province. In addition, the influence of the income and environmental cognition of farmers on the energy consumption of rural households around nature reserves was examined. The results showed that the regulatory policies of the nature reserves were conducive to reducing the traditional biomass energy consumption of the farmers. The nature reserves indirectly affect the energy consumption of the farmers by influencing their income, and cognition plays an important role in reducing the traditional biomass energy consumption of farmers in nature reserves. Compared with provincial nature reserves, national nature reserves have a more evident impact on the energy consumption of farmers. Based on the empirical results, farmers should be encouraged to reduce their high-pollution and high-emission energy consumption activities and should be provided with technical support and financial subsidies for clean energy, such as solar energy and biogas. The following measures should be taken to maintain the ecological environment of the nature reserves and to reduce the contradiction between the nature reserves and farmers: supervise the coal quality in the surrounding areas of nature reserves, improve the non-agricultural employment ability of farmers around nature reserves and the photovoltaic poverty alleviation project in Liaoning province, increase the income of farmers and promote the energy consumption of farmers around nature reserves, strengthen the management of provincial nature reserves, promptly change the traditional idea of ‘depend on the mountain and water’ adopted by farmers, improve the environmental awareness of farmers residing around nature reserves, and advocate green energy consumption.

## 1. Introduction

With the rapid economic growth and continuous consumption of resources and energy, ecological and environmental problems have become increasingly prominent [1,2]. China has taken a series of measures to solve these major problems and to protect the ecological environment and natural resources, among which the establishment of nature reserves is considered the best way to protect the ecosystem. The biological and ecological resources provided by nature reserves are not only strategic resources for the country but also the foundation of human survival and sustainable economic and social development [3]. However, because the space between the protected areas and the surrounding communities borders and overlaps and the resources are intertwined, the conflict between the protection of natural resources and the development of a community economy has become an unavoidable problem. Among the many contradictions between nature reserves and the surrounding communities, the energy consumption of farmers around nature reserves is an important factor affecting the coordinated development of nature reserves and the surrounding communities [4]. Villages around nature reserves are mostly located in remote mountainous areas, far away from cities, with limited living standards and economic conditions. Farmers mostly use local natural resources as their first choice of energy consumption [5]. With the establishment of nature reserves and the strengthening of protection measures, the traditional lifestyle of farmers residing around nature reserves and activities such as collecting herbs, planting, and logging, which directly depend on natural resources, are restricted, leading to the loss of some traditional energy sources. Therefore, promptly adjusting the production and lifestyle of residents around nature reserves and changing their energy consumption structure are important measures for the sustainable development of nature reserves.

Studies have shown that the energy consumption of farmers is mainly influenced by both internal and external factors associated with their families. With regard to the internal factors, their family income [6,7,8], the family structure [9,10], the educational qualification level of their family members [11,12], the householder’s age [13], and the non-agricultural employment of their family members [14] are the major factors that affect the energy consumption structure of farmers. In addition, as an individual in a social organization, the behavior and choices of a farmer are not only influenced by family and individual factors but also restricted by geographical areas and related policies [15]. Geographically, owing to the differences in resource endowments among different regions, the energy resources in the rural regions of Northeast China mainly include fuelwood and straw [16], while those in the rural regions of Hebei, Shandong, and Shanxi provinces mainly include coal and fuelwood [17]. In terms of policy, in 1998 China began to reform the rural power grid and rural power management system. To date, approximately 150 million farmers in 48,000 administrative villages have benefitted from the rural power network grid improvements, and the cost of electricity for farmers has decreased by 31.58% [18]. With the popularization of electricity, the rural energy consumption structure has fundamentally changed. To guide farmers to use renewable or clean biomass energy sources; accelerate energy construction in the rural areas of China; and promote the development of low-carbon clean energy sources such as biogas, wind energy, and solar energy, the Chinese government has issued a series of energy policies such as the 13th Five-Year Plan for Energy Development and the Action Plan for Energy Technology Revolution and Innovation, which guide or restrict the energy consumption behaviour of farmers by changing the external environmental conditions.

Furthermore, an understanding of low-carbon emission reductions and environmental pollution is key to promoting low-carbon consumption behaviour among farmers and improving the environmental quality. The higher the environmental awareness level of the farmers, the easier the adoption of low-carbon consumption behaviour [19]. Environmental awareness, as an important psychological factor for promoting the low-carbon and energy-saving behaviours of farmers, plays an important role in encouraging their families to reduce the use of high-emission and high-pollution energy sources [20]. The environmental cognition level of the farmers is influenced by many internal and external factors, such as their educational qualification, economic level, business scale, technical demand, risk preference, and government propaganda [21].

Because villages around nature reserves are mostly located in remote mountainous areas, their economy is relatively backward, meaning traffic is inconvenient. Therefore, farmers residing around nature reserves are highly dependent on natural resources. Several studies have reported on the influencing factors of rural energy around nature reserves, such as the internal factors associated with their families [22] and the regional external factors [4,23]. Few studies have explored whether there are differences in the energy consumption structures between farmers residing in nature reserves and those residing outside nature reserves. However, the following concerns remain unaddressed: Does the difference in the energy consumption structures result from the implementation of regulatory policies for nature reserves? Will the improvement of environmental awareness among farmers promote the impacts of regulatory policies in nature reserves on the energy consumption structure? Do nature reserves affect the energy consumption structure of farmers by affecting their total household income? Is there any difference in the degree of influence of different levels of nature reserves on the energy consumption structures of farmers? After the nature reserve suppresses the consumption of traditional energy such as firewood, how do farmers select their main alternative energy source?

In this study, a field investigation of farmers around the Laotu Dingzi, Baishi Lazi, and Haitangshan National Nature Reserves and the Monkey Rock, Sankuaishi, and Heshang Maozi Provincial Nature Reserves in Liaoning province is performed using the OLS regression and PSM methods. The aim is to explore the impacts of nature reserve policies on the energy consumption structure of the surrounding farmers, in order to deeply analyse whether the establishment of nature reserves has an impact on the consumption structure of the farmers, and to verify the mechanism of the impact of the moderating effect of the farmers’ environmental cognition and the mediating effect of the farmers’ household income on the energy consumption structure of the surrounding farmers. We also aim to put forward policy suggestions to promote the level of energy consumption of the farmers around the reserves and to advocate for green energy consumption.

## 2. Theoretical Analysis and Research Hypothesis

The Regulations of the People’s Republic of China on Nature Reserves stipulate that “no unit or individual is allowed to enter the core areas of nature reserves, and activities such as logging, grazing, and burning are prohibited in nature reserves” (Source: Regulations of the People’s Republic of China on Nature Reserves, http://www.gov.cn/, China Government Network, accessed on 10 October 2021). With the establishment of nature reserves, the forests originally managed by farmers are protected [24], and the behaviour of the farmers is restrained by such policies. A policy is mandatory. If the behaviours of farmers are found to violate policies and regulations, the government takes corresponding punishment measures. Owing to the regulatory constraints and punishments, the direct use of forest resources by farmers is restricted, the firewood collection cost to farmers is increased, and the firewood consumption rate of the farmers residing around nature reserves is greatly reduced. Based on this phenomenon, we propose the following hypothesis:

**Hypothesis** **1****(H1).** *Regulatory policies for nature reserves are conducive to reducing the traditional energy consumption rate of the farmers residing around nature reserves*.

According to the social behaviour theory, behavioural intention is the premise of an individual’s corresponding behavior, and there is a high consistency between the intention and behaviour [25]. If farmers are aware of resource wastage and environmental pollution, they will consciously reduce their use of traditional biomass energy sources and realise that renewable energy is eco-friendly and helps with resource utilization [20]. When farmers with different levels of environmental cognition encounter the same problem, their behaviours and responses will be different. Based on this phenomenon, we propose the following hypothesis:

**Hypothesis** **2** **(H2).***Increased environmental awareness will encourage farmers and their families to reduce their proportion of traditional energy consumption out of their total energy consumption*.

The energy ladder theory by Leach states that there are three types of household energy sources based on economic status (from low end to high end): traditional biomass energy sources such as firewood and straw; fossil energy sources such as coal, electricity, liquefied gas, and gasoline; and clean energy sources such as solar energy and biogas [26]. With an increase in income for the farmers, their households will first shift from traditional energy to fossil energy and eventually to clean energy, which is superior to both traditional and fossil energy sources. After the establishment of nature reserves, more employment opportunities become available for the local farmers, which changes their original livelihoods and increases their income level [27]. In addition, the government provides corresponding economic compensation to farmers whose production and living activities are restricted. Weiguang and Chang reported that with an increase in income, the amount of firewood used by farmers decreases, which indicates that the energy consumption of the farmers residing around nature reserves gradually changes [4]. The higher the income, the greater the possibility of the farmers using other energy sources. Based on this phenomenon, we propose the following hypothesis:

**Hypothesis** **3** **(H3).***The establishment of nature reserves indirectly reduces the traditional energy consumption rate of the neighbouring farmers by increasing their total income*.

Forest farmers who have lived in forest areas for generations are highly dependent on forest resources. This phenomenon is called the ‘knowledge-seeking environment’ [28,29]. Because the establishment time of national nature reserves is different than that of provincial nature reserves, the dependence of forest farmers on natural resources is different. The earlier the establishment of a nature reserve, the more beneficial it is to avoid a ‘knowledge-seeking environment’. In addition, the higher the administrative level of the nature reserves, the more standardised the management system and the stricter the punishment measures, which greatly inhibits the behaviours of farmers, such as for stealing and deforestation, thereby reducing their traditional energy consumption. Based on this phenomenon, we propose the following hypothesis:

**Hypothesis** **4** **(H4).***The higher the administrative level of the nature reserves, the more evident the impact on the energy consumption structure of the local farmers*.

The research framework is shown in Figure 1.

## 3. Description of Data, Variables, and Model Setting

### 3.1. Data Sources

This study was performed based on a questionnaire survey and an in-depth interview, and was conducted from June to July 2021. This study was financially supported by the National Social Science Fund ‘Research on the Impact of Collective Woodland Use Control in Nature Reserves on the Welfare of Surrounding Farmers and Ecological Compensation System’ (20BG173).

To make the samples more scientific and representative, a stratified random sampling method was used to prepare the questionnaire for the survey, and 3 national nature reserves and 3 provincial nature reserves were randomly selected from 19 national nature reserves and 27 provincial nature reserves in Liaoning province. Among the 16 towns (townships) located around the nature reserves, 2–3 villages within 15 km of the boundary of the nature reserves were selected from each township, and 20–25 families were selected from each village. A semi-structured interview, representing an important tool in a participatory rural appraisal (PRA) that is widely used at present, was used to query the farmers (Figure 2).

According to the boundaries of the nature reserves, the farmer families were divided into those residing in nature reserves and those residing outside nature reserves based on the survey. The production and life factors of the farmer families in the nature reserves were found to be directly or indirectly affected by the regulatory policies of the nature reserves, whereas those of the farmer families residing outside the nature reserves were not. The content of the questionnaire mainly included the following four aspects: (1) basic information regarding household heads, including the age, educational qualification, and political status of the household heads; (2) basic information regarding the farmer households, including the annual income, non-agricultural employment status, number of family members, production from animal husbandry, cultivated land area, and woodland area; (3) the household energy consumption factors for the farmers, including the type, quantity, source, and purpose of energy use; (4) the environmental awareness of the farmers residing around the nature reserves, including their awareness of low-carbon consumption, environmental protection, and environmental policies related to the nature reserves.

A total of 44 villages in 16 towns (townships) in 6 counties where the 6 nature reserves are located were selected for the questionnaire survey. A total of 1002 questionnaires were distributed and collected, of which 46 invalid questionnaires were excluded and 956 valid questionnaires were retained, with an effective response rate of 95.4% (Table 1).

### 3.2. Description of Variables

#### 3.2.1. Explained Variable: Household Energy Consumption Structure of Farmers

In 2020, 509 million permanent residents were reported to reside in rural areas in China (Data source: National Bureau of Statistics http://www.stats.gov.cn/, population census rural population (2020), accessed on 10 October 2021). The energy consumption structure of the rural households is an important monitoring index for rural energy infrastructure construction and the living standards, indoor air quality, and health levels of the rural residents. In this study, the ratio of traditional biomass energy consumption (firewood and straw) to the total energy consumption (firewood, straw, electricity, coal, liquefied gas, gasoline, solar energy, and biogas) was used to measure the household energy consumption structure of the farmers, whereby the higher the ratio, the higher the household consumption of firewood and straw. 

#### 3.2.2. Explanatory Variable: Are the Villages of Farmer Families Affected by the Regulatory Policies of the Nature Reserves?

As individuals in social organisations, the choices of farmers are not only influenced by familial internal factors but also restricted by familial external factors. Government policies often guide and restrict household energy consumption rates. In this study, the virtual variables ‘0’ and ‘1’ were used to examine whether the villages of farmer families were affected by the regulatory policies of the nature reserves. The boundaries of the nature reserves were considered the dividing lines. If a village located in a nature reserve was directly or indirectly affected by the regulatory policies, the variable assigned was 1; otherwise, the variable assigned was 0. The negative regression coefficients indicated that the regulatory policies reduced the traditional biomass energy consumption of the farmers residing around the nature reserves, and vice versa.

#### 3.2.3. Intermediate Variable: Total Annual Income of Farmer Families

According to the energy ladder theory, the energy consumption structure of the farmers residing around nature reserves is not only directly affected by the regulatory policies of the nature reserves but also indirectly affected by the total annual income of the farmers. With an increase in income, the farmers transition from traditional energy to fossil or clean energy sources. In this study, the total annual income of the farmer families was used as an intermediate variable to analyse the effects of the total annual income of the farmer families on the impact of the regulatory policies of the nature reserves on the energy consumption structure.

#### 3.2.4. Regulatory Variable: Environmental Cognition of Farmers

As a type of comprehensive cognition, the environmental cognition of the farmers has no specific index to measure its level [30]; therefore, it is necessary to construct a relevant index system in order to make a comprehensive judgment. Because there are many relationships among multiple indicators, and as the values of indicators are not in the same dimension, it is difficult to compare them directly. In this study, the entropy method was used to objectively measure the environmental cognition of the farmers based on their awareness regarding the low-carbon consumption, environmental protection, and environment-related policies of the nature reserves. The results are shown in Table 2.

#### 3.2.5. Control Variables

In addition to the regulatory policies on the nature reserves and the total household income, both internal familial factors and external geographic conditions influence the energy consumption structures of farmers. In this study, the age, educational qualification level, and political status of the head of the household; the number of members involved in non-agricultural employment; the size of the family; the number of agricultural machinery pieces; the number of large household appliances; the production from animal husbandry; the area of arable land; the area of forest land; the number of capable people among their relatives and friends; and the relationships between human expenditure and the degree of transportation convenience were used as control variables to scientifically and accurately measure the impacts of the regulatory policies of the nature reserves on the energy consumption of the farmer households. 

The descriptive statistics for the abovementioned variables, definitions, and related variables are listed in Table 3.

### 3.3. Model Establishment

#### 3.3.1. Benchmark Estimation Model

To investigate the impacts of the regulatory policies of the nature reserves on the energy consumption structure of the farmers, the following linear regression equation was used:(1)$$Y_{j}=\beta_{0}+\beta_{1}T+\overset{n}{\underset{i=2}{\sum }}\beta_{i}Control_{ij}+\varepsilon_{i}$$

In Equation (1), the explanatory variable $Y_{j}$ indicates the energy consumption structure of the farmers; the subscript j indicates the j farmer families; the explanatory variable T indicates whether the village of farmer families is affected by the regulatory policies of nature reserves; Control represents a series of control variables; $\beta_{0}$, $\beta_{1}$, and $\beta_{i}$ are the coefficients to be estimated and $\varepsilon_{i}$ is a random error term.

#### 3.3.2. Intermediate Effect Model

To verify the intermediate effects of the regulatory policies of the nature reserves on the total household income of the farmers, the intermediate effect test model proposed by Zhonglin et al. [31] was used to formulate the following equations:(2)$$Y=\beta_{1}+\kappa T+\mu_{1}$$
(3)$$M=\beta_{2}+\gamma T+\mu_{2}$$
(4)$$Y=\beta_{3}+\theta T+\omega M+\mu_{3}$$

In the abovementioned equations, the explained variable Y represents the energy consumption structure of farmers; the explanatory variable T represents whether the village of farmer families is affected by the regulatory policies; the intermediate variable M represents the total income of farmer families; $\beta_{1}$, $\beta_{2}$, and $\beta_{3}$ represent the intercept; $\mu_{1}$, $\mu_{2}$, and $\mu_{3}$ are random disturbance terms; and κ, γ, θ, and ω are the parameters to be estimated.

#### 3.3.3. Moderating Effect Model

Because the environmental cognition of the farmers is closely related to their energy consumption, the nature reserve policies may regulate their energy consumption structure. Based on the interaction (T×Cognition) between the nature reserve policies and environmental cognition, the improved equation is as follows:(5)$$Y_{j}=\beta_{0}+\beta_{1}T+\beta_{2}T\times Cognition+\overset{n}{\underset{i=3}{\sum }}\beta_{i}Control_{ij}+\varepsilon_{i}$$

Considering that the introduction of interactive items may lead to multicollinearity, the abovementioned formula centralises the interactive items.

#### 3.3.4. Propensity Score Matching

Considering that factors other than the regulatory policies of the nature reserves can affect the household energy consumption, propensity score matching (PSM) was used to examine the net effect of the nature reserves on the household energy consumption structure. Using only ordinary least squares (OLS) for the regression analysis may result in deviations. Therefore, to eliminate errors caused by other factors, PSM was used. The farmers residing in the nature reserves were included in the experimental group, whereas those residing outside the nature reserves were included in the control group. Subsequently, a ‘counterfactual framework’ was constructed to control the influence of other factors so that the results were more reliable.

First, the tendency score was estimated via logistic transformation:(6)$$p\left(X_{i}\right)=p_{r}\left[D=1|X_{i}\right]=\frac{exp\left(\beta X_{i}\right)}{1+esp\left(\beta X_{i}\right)}$$

In Equation (6), D is a processing variable (if the farmer family resides in the nature reserve, the value is 1; if the farmer family resides outside the nature reserve, the value is 0) and $X_{i}$ is a covariate, such as the total household income, the age of the head of the household, the number of large household appliances, the woodland area, and the degree of transportation convenience; p is the tendency score to be estimated.

Furthermore, the tendency scores of the farmers residing in and outside the nature reserves were matched via PSM. Radius matching, K-nearest neighbour matching, and kernel matching were used, and the results were compared. If the results were similar, the matching was considered robust.

Finally, the average treatment effect on the treated (ATT) was estimated, and the impact of the regulatory policies of the nature reserves on the energy consumption structure of the rural households was examined:(7)$$ATT=E\left[Y_{1}^{i}-Y_{0}^{i}\right]=E\left[Y_{1}^{i}-Y_{0}^{i}|D=1\right]=E\left[Y_{1}^{i}|D=1\right]-E\left[Y_{0}^{i}|D=1\right]$$

In Equation (7), $Y_{0}^{i}$ represents the potential results of two counterfactual situations (residence inside and outside the nature reserve), $E\left[Y_{1}^{i}|D=1\right]$ is the expected rural household energy consumption structure of the families residing in the nature reserves, and $E\left[Y_{0}^{i}|D=1\right]$ is the expected rural household energy consumption structure of the families residing outside the nature reserves. The difference between $E\left[Y_{1}^{i}|D=1\right]$ and $E\left[Y_{0}^{i}|D=1\right]$ is the effect of the nature reserve policy on the energy consumption structure of the farmers.

## 4. Descriptive Statistics and Analysis

According to the field survey data, the household energy sources of the farmers residing around the Liaoning nature reserves mainly included firewood, straw, coal, electricity, liquefied gas, gasoline, solar energy, and biogas. Through a comparative analysis of the energy sources and ecological environment effects, the eight types of energy sources were divided into the following three categories: traditional biomass energy, fossil energy, and clean energy [32,33]. The traditional biomass energy sources mainly included firewood and straw; the fossil energy sources mainly included coal, electricity, liquefied gas, and gasoline; and the clean energy sources mainly included solar energy and biogas. For our convenience in the comparative analysis, the units of different types of energy were converted into standard coal as the unified calculation unit according to the China Energy Statistics Yearbook 2020—Reference Coefficient for Converting Various Energy Sources into Standard Coal:(8)$$E_{i}=\overset{n}{\underset{i=1}{\sum }}B_{i}\times C_{i}$$

In Equation (8), $E_{i}$ is the total amount of standard coal converted from the i energy source, $B_{i}$ is the total amount of the original unit of the i energy source, and $C_{i}$ is the conversion standard coal coefficient. The conversion standard coal coefficient values (kg standard coal/kg) for raw coal, gasoline, liquefied gas, corn straw, firewood, and biogas were 0.7143, 1.4714, 1.7143, 0.529, 0.571, and 0.714, respectively. Because no fixed conversion method was available for solar energy, the energy consumption for solar water heaters was estimated based on the calculation of the heat consumption owing to increasing water temperatures, which was converted into the standard coal consumed versus solar energy [34]:(9)$$T=c\times m_{i}\times \Delta t_{i}\times C$$

In Equation (9), T is the energy consumption of the solar water heaters with standard coal as the unit, is the specific heat capacity of water ($4.2\times 10^{3}\;J/\left(\mathrm{kg}\cdot {}^{\circ}C\right)$), $m_{i}$ is the water storage capacity of the different types of solar water heaters, $\Delta t_{i}$ is the temperature difference of the water in different seasons, and C is the standard coal coefficient of water heat conversion (0.03412 kg standard coal/million J). The results are shown in Figure 3.

### 4.1. Traditional Energy Consumption

The direct combustion of traditional biomass energy has always been one of the most important ways to obtain rural energy. In this study, the traditional energy consumption rates of farmers residing in and outside nature reserves accounted for 73.83% and 60.30% of the total energy consumption, respectively. Because the villages with farmers residing around nature reserves are located in the forest area and the forest resources are the important means of production and livelihood for the farmers, firewood is the most important energy source except electricity. The household energy consumption rates of the farmers residing outside and inside nature reserves accounted for 47.92% and 34.59% of the total, respectively, while the firewood consumption inside the nature reserves was reduced by 902.61 kgce compared with that outside the nature reserves. The on-site investigation revealed that the consumption of straw was mainly affected by the cultivated land area used by the farmer families. The average cultivated land area sizes of the farmer families residing inside and outside nature reserves were similar; therefore, no difference was observed in straw consumption between the farmer families residing inside and outside the nature reserves.

### 4.2. Fossil Energy Consumption

With the improvement of the petrochemical industry and the living standards of the general population, people have paid more attention to convenience and sanitation regarding energy use [23], and fossil energy has become indispensable to the rural energy consumption structure. According to the survey, the average household consumption rates of fossil energy inside and outside the nature reserves were 1608.218 kgce and 1167.375 kgce, respectively. Being influenced by the related policies of the nature reserves, such as cutting restrictions and the closing of hillsides to facilitate afforestation, most farmers in the area replace firewood with coal in winter. As a result, the coal consumption rate of the farmers residing in nature reserves has reached 1097.925 kgce, which is approximately 30% higher than that of the farmers residing outside the nature reserves.

### 4.3. Clean Energy Consumption

Clean energy is an emerging energy resource with several advantages related to sustainable production and reducing environmental pollution [32]. According to the survey, the average household solar energy consumption rate of the farmer families residing in the nature reserves was 169.001 kgce, accounting for 3.76% of the total energy consumption, whereas that of the farmers residing outside the nature reserves was 173.172 kgce, accounting for 3.39% of the total energy consumption. The average biogas consumption rate of the farmers residing in Liaoning was 6.323 kgce, whereas that of the farmers residing outside Liaoning was 1.392 kgce, accounting for the lowest energy consumption rate. The on-field investigation revealed that the average annual temperature range in Liaoning is between 7 °C and 11 °C, with colder temperatures in winter. The temperature can reach −30 °C (Liaoning Provincial People’s Government Network http://www.ln.gov.cn/, accessed on 10 October 2021), which is not conducive to the fermentation of biogas digesters [35], decreases the utilisation rate of biogas digesters, and fails to adequately contribute to the use of other biogas digesters.

## 5. Empirical Results and Analysis

Stata (version 15.0) software (Runze Li, Shenyang, China. Purchase and download from the Stata website) was used to examine the impact of the regulatory policies related to nature reserves on the energy consumption rates of farmers. First, based on Equation (1), OLS regression was used to examine the impact of the nature reserve policies on the energy consumption structure of the farmers, and the control variables were added to the regression model in (2). Second, the farmers residing inside and outside the nature reserves were divided into the experimental and control groups via PSM, and the ‘net effect’ of the nature reserve policies on the energy consumption structure of the farmers was analysed. Furthermore, to analyse the regulatory effects of environmental cognition on the impact of the nature reserve policies on the energy consumption structure of the farmers, an interaction model in (3) of variables affected by the nature reserve policies and the environmental cognition of the farmers was introduced. The intermediate effect model was used to test the models in Equations (4) and (5), which suggested that the establishment of nature reserves indirectly affects the traditional energy consumption by providing employment opportunities and increasing the income level of local farmers. Seemingly unrelated regression (SUR) was used to evaluate the coefficient differences between groups to examine differences in the impacts of national and provincial nature reserves on the energy consumption structure of the local farmers via Equations (6) and (7).

Before using the formal regression analysis, the multicollinearity of each variable was assessed. The results showed that the variance inflation factor (VIF) of each variable was <2, with an average VIF of 1.22, indicating that there was no serious collinearity problem. The regression results are shown in Table 4.

### 5.1. Benchmark Regression and Adjustment Effect Analysis

According to the results of Equation (1), the nature reserves had a significant negative impact on the household energy consumption structure of the farmers. This negative impact was also significant at the level of 1% in Equation (2) after adding the control variables, with a regression coefficient of −0.075, which indicated that the regulatory policies of nature reserves are conducive to reducing the household traditional biomass energy consumption rates of farmers. Therefore, Hypothesis 1 was verified.

Equation (5) was used to examine the moderating effects of environmental cognition on the impact of the nature reserve policies on the energy consumption structure of the farmers. The results are shown in Table 4. In (3), the cross-correlation regression coefficient was −0.035, which was significant at the level of 1%, indicating that the environmental cognition of the farmers significantly affected the impact of the nature reserves on the energy consumption structure of the farmers. These results suggested that environmental cognition plays a catalytic role in reducing the household traditional biomass energy consumption of farmers residing in nature reserves. Therefore, Hypothesis 2 was validated.

In terms of the control variables, the age of the head of the household has a significant positive impact on the ratio of traditional-to-total energy consumption; that is, with an increase in the householder’s age, the consumption of traditional energy increases. The younger the head of a household, the easier it is to avoid the ‘knowledge-seeking environment’. The educational qualification of household heads has a significant negative impact on the ratio of traditional-to-total energy consumption. Compared with household heads with moderate and lower educational qualifications, well-educated household heads are more likely to accept clean energy sources such as solar energy and biogas, thereby reducing the consumption of traditional energy [36]. Furthermore, non-agricultural employment has a significant negative impact on the ratio of traditional-to-total energy consumption. First, non-agricultural employment can significantly increase the household income of farmers, indicating that farmers can buy fossil energy at a higher cost. Second, farmers who go out to work are more likely to explore new things and avoid the ‘knowledge-seeking environment’, thereby reducing the consumption of traditional energy. The family size has a significant positive impact on the ratio of traditional-to-total energy consumption. The larger the family size, the greater the energy demand. Because firewood is the main energy source for forest farmers, its consumption rate is higher, leading to an increase in the consumption of traditional energy. Large household appliances have a significant negative impact on the ratio of traditional-to-total energy consumption. With an increase in the use of household appliances, the electricity consumption also increases, leading to a decrease in the consumption of traditional energy. In addition, the popularity of electric cookers, water heaters, microwave ovens, and other appliances moderately reduces the use of traditional cookware. The forestland area owned by farmers has a significant positive impact on the ratio of traditional-to-total energy consumption. The larger the forestland area, the lower the collection cost for farmers and the higher the number of farmers using traditional energy sources such as firewood. The political status of the friends and relatives of the farmer families and the total expenditure of a family for attending weddings and funerals in 1 year have significant negative effects on the ratio of traditional-to-total energy consumption. The higher the number of people with political status, the higher the understanding of national policies among farmers and the higher the use of clean and low-carbon energy sources [37]. The transportation convenience has a significant negative impact on the energy consumption structure of farmer families. The higher the transportation convenience of farmers in residential areas, the more favourable it is to purchase fossil energy sources such as coal and liquefied petroleum gas. In addition, it promotes the use of clean energy sources such as solar energy.

### 5.2. PSM

To eliminate the estimation bias caused by factors such as the characteristics and consumption preferences of the farmers, PSM was used to verify the impact of the regulatory policies of nature reserves on the energy consumption structure of the farmers. Based on Table 4, the control variables were matched between the control and experimental groups, and the farmers residing in the nature reserves were included in the experimental group, whereas those residing outside nature reserves were included in the control group. Furthermore, K-nearest neighbour matching, kernel matching, and radius matching were used to estimate the ATT of the energy consumption structure of the rural households.

#### 5.2.1. Matching Test of the PSM Model

A balance test was performed before the matching analysis, and the results are shown in Figure 4. No significant difference was observed in the control variables after matching. In addition, considering the K-nearest neighbour matching method as an example, the distribution of the kernel density function of the propensity scores was examined in the experimental (farmers residing in nature reserves) and control (farmers residing outside nature reserves) groups before and after matching (Figure 5). The kernel density functions of the experimental and control groups were quite different before matching; however, the difference between them was decreased after matching. Moreover, the trend was the same and the matching effect was good, indicating that the PSM model was suitable for analysis.

#### 5.2.2. ATT Analysis

The results of the PSM (Table 5) revealed that irrespective of the use of the K-nearest neighbour or kernel matching method, the regulatory policies of the nature reserves significantly reduced the traditional energy consumption of the farmers at the confidence level of 1%, indicating that the policies had a significant negative impact on the energy consumption structure of the farmers. Therefore, these results verified Hypothesis 1.

### 5.3. Analysis of Mediatory Effects

The mediation effect analysis model proposed by Zhonglin et al. [31] was used to examine the mediatory effects of the total income of the farmers on the impact of nature reserve policies on the energy consumption structure of the farmers. The regression results are shown in Table 6.

First, considering the total household income of the farmers as the explanatory variable, the influence of the nature reserve policies on the total household income of the farmers was observed. The results showed that the regulatory policies of nature reserves had a significant positive impact on the total household income of the farmers at a significance level of 1%. This positive correlation may be attributed to the establishment of nature reserves, which increases the employment opportunities for the local farmers. In addition, the government provides corresponding economic compensation to farmers whose production and living activities are restricted, thereby increasing the income level of the farmers. Second, based on the intermediate effect analysis, the energy consumption structure of the farmers was considered an explanatory variable, and the income of the farmer families and the impact of the nature reserve policies were included in the model. The results showed that both the nature reserve policies and the total income of the farmers had significant negative impacts on the energy consumption structure of the farmers. Combined with the results from (2), the results suggested that the total household income of the farmers played a partial intermediate role in the relationship between the nature reserve policies and the energy consumption structure of the farmers. This phenomenon may be attributed to the energy ladder theory. In addition to the direct influence of the nature reserve policies, the income of the farmers is another important factor affecting the energy structure of the farmers [26]. The higher the household income of the farmers, the higher the inclination toward using fossil energy and clean energy, thereby reducing the consumption of traditional energy such as firewood [38]. These results verified Hypothesis 3.

### 5.4. Heterogeneity Analysis

Owing to the differences in the establishment periods and management of nature reserves, different administrative levels of nature reserves have different effects on the energy consumption structure of the farmers. In this study, the farmers were divided into those residing around national nature reserves and those residing around provincial nature reserves. Based on the OLS model, the two groups were examined using SUR, the significance of the coefficient differences between the groups was examined, and the differences between the national (in (6)) and provincial (in (7)) nature reserves were discussed.

As shown in Table 7, the *p*-value of the SUR was 0.002, which was significant at the level of 1%. The regression coefficients of the two groups were compared via SUR. The results of the OLS regression revealed that both the national and provincial nature reserves had significant impacts on the energy consumption structure of the farmers. The regression coefficient was negative; however, the regression coefficient of the national nature reserves (−0.121) was significantly higher than that of the provincial nature reserves (−0.035). These results indicate that farmers residing around national nature reserves are more sensitive to the regulatory policies of national nature reserves than to those of provincial nature reserves, and national nature reserves have a greater influence on the energy structures of local farmer families. Therefore, Hypothesis 4 was verified.

## 6. Discussion

Research into the energy consumption structures of rural households around nature reserves is important in reducing the energy consumption of rural households around nature reserves, promoting green energy consumption, reducing the consumption of high-emission and high-pollution energy sources for rural households around nature reserves, and alleviating the conflicts between rural households and the establishment of nature reserves.

Based on the study by Qiu, eight types of energy sources were divided into three categories as follows: traditional biomass energy, fossil energy, and clean energy [32]. The energy consumption for solar water heaters was estimated using the ‘China Energy Statistics Yearbook 2020—Reference Coefficient for Converting All Kinds of Energy into Standard Coal’, based on the heat consumption for increasing water temperatures. In addition, different types of energy were converted into standard coal as the quantitative unit, which provides a reference basis and method for exploring energy consumption structures.

The total household income of farmers plays a partial intermediate role in the relationship between nature reserves and the energy consumption structure of farmers. This finding is consistent with that of a previous study [24] and reveals the effects of nature reserves on the energy consumption structure of farmers residing around them. Regulatory policies of nature reserves restrict the behaviours of farmers, which increases the cost of traditional energy consumption for farmers. In addition, nature reserves provide more employment opportunities for local farmers and the government provides corresponding economic compensation to farmers who restrict their production and living activities, thus increasing the income level of farmers and the consumption of fossil and clean energy among farmers.

The educational qualification level of the head of the household, the number of members engaged in non-agricultural employment, the number of large household appliances, the number of people with political status among relatives and friends, the expenditure on human relations, and the degree of transportation convenience had significant negative effects on the ratio of traditional-to-total energy consumption. However, the age, household size, and woodland area had significant positive effects on the ratio of traditional-to-total energy consumption, which was consistent with the results of previous studies by Emodi and Weiguang [4,10]. However, the number of livestock animals used for breeding had no significant impact on the energy consumption structure of the farmers, which was inconsistent with the results of the abovementioned studies. This inconsistency may be attributed to the small number of farmers involved in livestock farming, meaning it does not significantly impact the energy consumption structure of the farmers.

This study attempted to answer the following important question: Do nature reserves affect the energy consumption structure of the local farmers? However, this study had some limitations. First, northern farmers, who mainly consume firewood and coal, were selected for analysis. Farmers in other areas may use different energy sources and have different habits, warranting further investigation. Second, no comparison was made between the results obtained in the course of the study and the global practice in the field. The third, the effects of changes in the energy consumption structure of local farmers owing to the regulatory policies of the nature reserves on the environment were not extensively discussed. By estimating the emissions of CO_2_ and major pollutants generated via the energy consumption of farmers, the use of high-emission and high-pollution energy sources can be reduced to facilitate the sustainable development of nature reserves, improve the quality of life of the farmers, and reduce health risks [39].

To improve the income level of the farmers residing around nature reserves, advocate green energy consumption among farmers, and strengthen the management of nature reserves, we propose the following suggestions based on the survey: First, establish a diversified energy consumption system, encourage farmers to reduce their consumption of high-pollution and high-emission energy sources, provide technical support and financial subsidies for the use clean energy such as solar energy and biogas, and reduce the consumption burden of farmers and the contradiction between nature reserves and local farmers. Second, reduce coal consumption in households, increase the research and development of alternative technologies to fossil energy such as coal, develop new and clean energy technologies, reduce the use cost of clean energy, and gradually replace traditional fossil energy with clean energy and renewable energy sources. Third, make complete use of the photovoltaic poverty alleviation project in Liaoning province, build small photovoltaic power stations in villages, increase the income of poor people, and improve the purchasing power for high-quality energy for farmers. Fourth, strengthen the non-agricultural employment training of farmers around nature reserves, improve the non-agricultural employment ability of farmers around nature reserves, effectively increase the income of farmers, and encourage farmers to transform the energy consumption structure. Fifth, improve the environmental awareness among farmers residing around nature reserves, advocate for green energy consumption, and increase the clean energy consumption. Sixth, strengthen the management of provincial nature reserves and promptly change the traditional idea of ‘depending on the mountain and the water’ among farmers.

## 7. Conclusions

Based on the survey data from 956 farmer households in six nature reserves in Liaoning province, the impacts of nature reserve policies on the energy consumption structure of the farmers was examined empirically. The main conclusions were as follows.

In terms of traditional energy, the firewood consumption rate of the farmers residing in the nature reserves was lower (902.61 kgce) than that of the farmers residing outside the nature reserves because the activities of the farmers residing in the nature reserves are restricted by the regulatory policies of these nature reserves. In terms of fossil energy, the average household consumption rates of fossil energy inside and outside the nature reserve were 1608.218 kgce and 1167.375 kgce, respectively. Influenced by the regulatory policies of the nature reserves, such as cutting restrictions and the closing of hillsides for afforestation, most farmers residing in these nature reserves replace firewood with coal during winter, thereby increasing their coal consumption (an approximate increase of 30% compared with the coal consumption outside the nature reserves). In terms of clean energy, no differences were observed in the solar energy consumption rates between the farmers residing inside and outside the nature reserves, and the consumption of biogas was the lowest. The on-field investigation revealed that low temperatures in Liaoning can reach −30 °C, which is not conducive to the fermentation of biogas digesters. Therefore, the utilisation rate of biogas digesters is low. Moreover, this means that biogas digesters cannot replace other energy sources.

Nature reserves have a significant negative impact on the energy consumption structure of farmers; that is, the regulatory policies of nature reserves are conducive to reducing the consumption of traditional biomass energy among farmers. These policies isolate local farmers via administrative means, prohibit the cutting and use of all resources from mountain forests, and reduce the traditional energy consumption of the farmers.

Environmental cognition, as a regulatory variable, plays a promoting role in reducing the traditional biomass energy consumption of farmers. As an important psychological motivating factor for farmers to reduce their carbon emissions, environmental cognition plays an important role in reducing the consumption of high-emission and high-pollution energy sources and promoting the use of eco-friendly energy [20].

The total income of farmer families plays an intermediate role in the relationship between the nature reserve policies and the energy consumption structure of the farmers. According to the energy ladder theory, with an increase in income, the farmers will shift from traditional energy to transitional energy and eventually to clean energy, which is superior to both traditional and fossil energy sources.

Nature reserves show some heterogeneity in influencing the traditional biomass energy consumption of farmers. Compared with farmers residing around provincial nature reserves, those residing around national nature reserves are more sensitive to regulatory policies. National nature reserves have been established for a long time, which has reduced the high dependence of farmers on forest resources to a certain extent [28]. In addition, the management of national nature reserves is stronger, and the corresponding punishment measures are stricter, which effectively inhibits both deforestation and forest theft.

## Figures and Tables

**Figure 1 ijerph-19-11955-f001:**
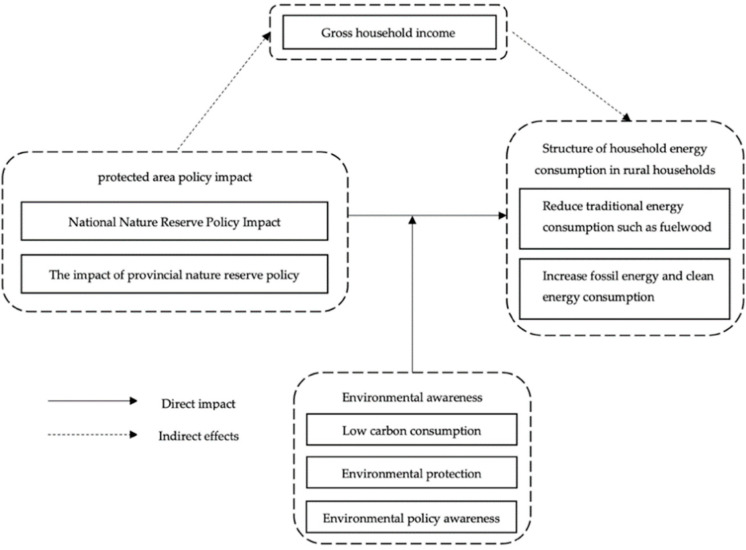
The research framework.

**Figure 2 ijerph-19-11955-f002:**
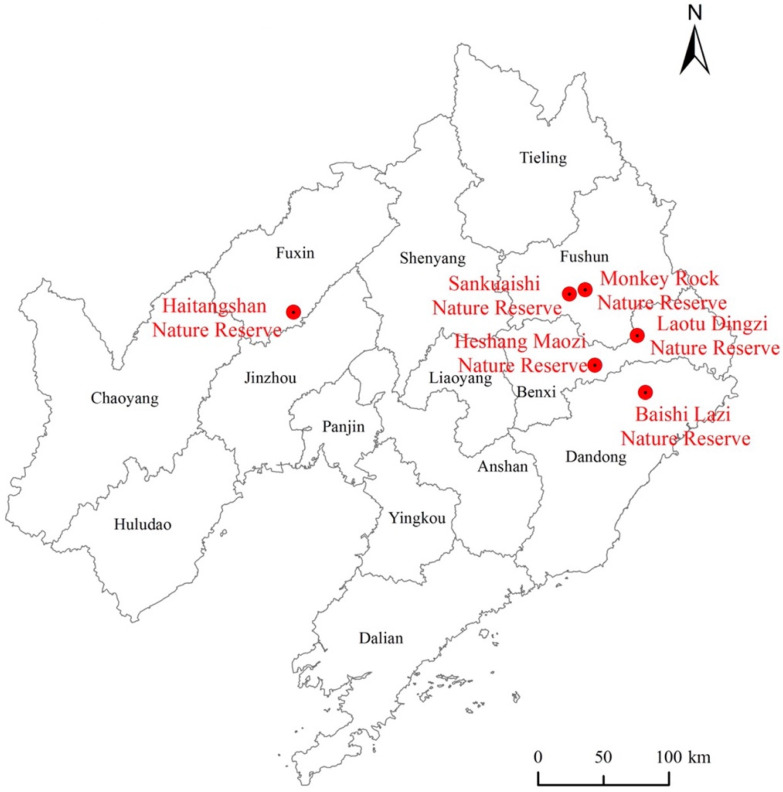
Illustration of the research area.

**Figure 3 ijerph-19-11955-f003:**
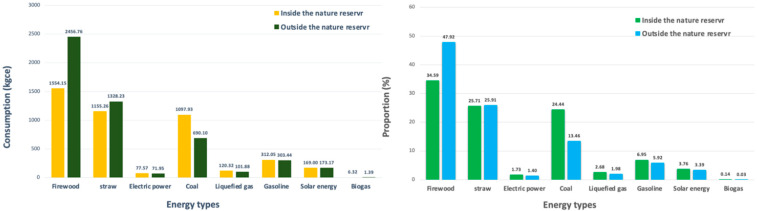
Energy consumption (average quantity) rates of rural households in and outside nature reserves.

**Figure 4 ijerph-19-11955-f004:**
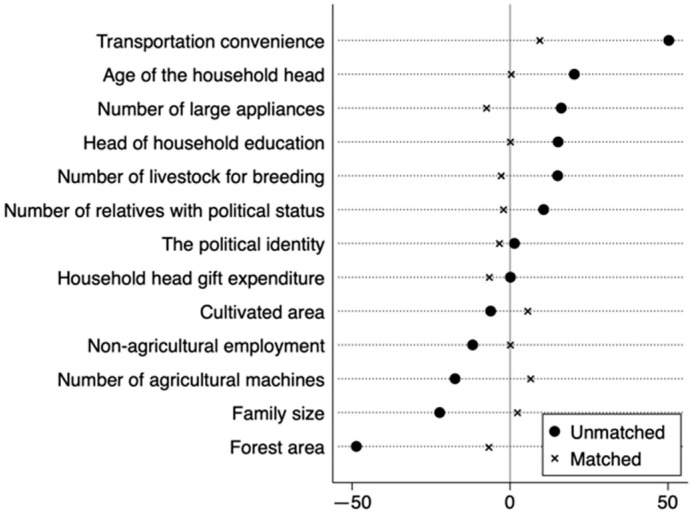
The deviation distribution of the sample covariates.

**Figure 5 ijerph-19-11955-f005:**
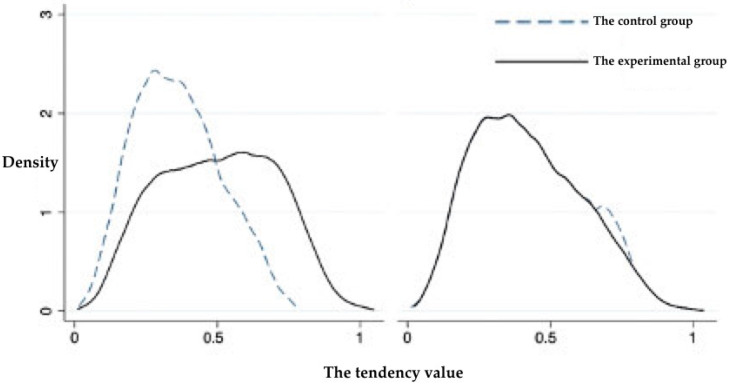
Matching of the kernel densities between the experimental and control groups.

**Table 1 ijerph-19-11955-t001:** Administrative levels and establishment times of the 6 nature reserves in Liaoning province and the numbers of valid questionnaires.

Name of the Nature Reserve	Administrative Level	Establishment Year	Number of Villages Surveyed (Case)	Number of Valid Questionnaires (Copies)
Laotu Dingzi Nature Reserve	National level	1981	9	184
Baishi Lazi Nature Reserve	National level	1981	4	75
Haitangshan Nature Reserve	National level	1986	14	309
Monkey Rock Nature Reserve	Provincial level	2003	5	118
Sankuaishi Nature Reserve	Provincial level	2003	8	194
Heshang Maozi Nature Reserve	Provincial level	2005	4	76
Total			44	956

**Table 2 ijerph-19-11955-t002:** Entropy coefficient values for the environmental cognition of the farmers.

Questions	Information Entropy	Value of Information Utility	Value of Weight Coefficient (%)
1. Do you understand the composition of greenhouse gases?	0.973	0.027	18.40
2. Do you think burning straw will cause environmental problems such as the greenhouse effect and air pollution?	0.990	0.010	6.76
3. Do you know that coal is a non-renewable resource?	0.982	0.018	12.00
4. Do you think poor air quality and water pollution have an impact on your health?	0.980	0.019	13.37
5. When the nature reserve was established, did you know about it?	0.984	0.016	10.95
6. Do you know about the control of collective forest land use in nature reserves?	0.984	0.016	11.10
7. Do you know that your production and operation in nature reserves require you to comply with national environmental protection standards?	0.979	0.021	14.27
8. Do you think the establishment of nature reserves is helpful for environmental protection?	0.981	0.019	13.15

**Table 3 ijerph-19-11955-t003:** The definitions of the variables and the descriptive statistics (mean).

Variable	Meaning	Whole Cohort (*N* = 956)	Farmers Residing Within the Protected Area (*N* = 402)	Farmers Residing Outside the Protected Area (*N* = 554)
Household energy consumption consumption structure	The ratio of traditional biomass energy to total energy	0.742	0.673	0.792
Policy implications for nature reserves	Is energy consumption restricted by the policies of nature reserves? (yes, 1; no, 0)		1	0
Total household income	Annual gross income of family members (yuan)	73,010.09	74,871.01	71,149.17
Environmental cognition of farmers	Awareness of farmers regarding low-carbon consumption, environmental protection, and environment-related policies of nature reserves (calculated using the entropy method)	2.97	3.20	2.80
Age of the household head	Age of the head of the household during the survey (years)	45	46	44
Educational qualification of the household head	Educational qualification of the household head *	2.88	2.95	2.84
The political identity of the head of the household	Whether the household head is or was a village official or a Communist Party member (yes, 1; no, 0)	1.321	1.326	1.317
Non-agricultural employment	Number of family members engaged in non-agricultural occupations (person)	1.339	1.276	1.384
Family size	Number of people living at home for >6 months in 1 year (person)	2.77	2.62	2.88
Number of agricultural machines	The number of agricultural machines used in the past year in the farm (sets)	0.80	0.72	0.85
Number of large appliances	Number of large household appliances (sets)	8.17	8.56	7.89
Number of livestock used for breeding	Quantity of livestock used for breeding (head)	0.20	0.40	0.06
cultivated area	The total area of cultivated land owned by the family (m^2^)	8200	7933.33	8393.33
Forest area	The total area of forest land owned by the family (m^2^)	46,833.33	46,033.33	47,413.33
Number of capable people among relatives and friends	The number of people with political status among family and friends that the head of the household moves around frequently (person)	1.22	1.32	1.14
Household head gift expenditure	The total household expenditure on marriage and funeral expenses in a year (yuan)	12,100.96	12,485.85	11,821.67
Degree of convenience of transportation	Distance from village to nearest provincial road (km)	11.44	13.85	9.69

Note: * Educational qualification level of the head of the household: 1 = none; 2 = primary school; 3 = junior high school; 4 = high school/secondary school; 5 = university college; 6 = undergraduate; 7 = postgraduate (including masters and doctoral degrees).

**Table 4 ijerph-19-11955-t004:** OLS regression results.

Variable	Model (1)	Model (2)	Model (3)
Policy implications for nature reserves	−0.119 *** (−8.62)	−0.075 *** (−6.135)	−0.044 *** (−4.803)
Environmental cognition of farmers			−0.123 *** (−23.297)
Policy implications for nature reserves × Environmental cognition of farmers			−0.035 *** (−3.815)
Age of the head of the household		0.002 *** (3.279)	0.001 (1.239)
Educational qualification of the head of the household		−0.017 ** (−2.146)	0.005 (0.832)
The political identity of the head of the household		0.012 (1.260)	0.010 (1.391)
Non-agricultural employment		−0.021 *** (−3.326)	−0.006 (−1.092)
Family size		0.058 *** (12.393)	0.033 *** (8.188)
Number of agricultural machines		0.002 (0.313)	−0.004 (−0.611)
Number of large appliances		−0.009 *** (−5.779)	−0.004 *** (−3.590)
Number of livestock for breeding		0.0001 (0.113)	0.002 (0.947)
Cultivated area		0.002 (0.307)	0.003 (0.549)
Forest area		0.014 *** (4.629)	0.010 *** (4.373)
Number of people with political identities among family and friends		−0.006 * (−1.732)	−0.005 ** (−2.008)
Household head gift expenditure		−0.018 ** (−2.335)	−0.006 (−0.926)
Transportation convenience		−0.043 *** (−4.847)	−0.022 *** (−3.135)
Constant	0.792 *** (15.465)	1.581 *** (15.465)	1.254 *** (15.757)
Sample size	956	956	956
R²	0.077	0.363	0.641
Adjusted R²	0.076	0.353	0.634

Note: (1) *, **, and *** represent significance at the levels of 10%, 5%, and 1%, respectively; (2) T values are mentioned in parentheses; (3) to eliminate the influence of heteroscedasticity, numerical variables such as the total household income, cultivated land area, woodland area, household head gift expenditure, and the degree of transportation convenient convenience were transformed via logarithm.

**Table 5 ijerph-19-11955-t005:** Processing of the PSM method.

Matching Method	Sample	Treatment Group	Control Group	D-Value	Standard Error	T Value
Before matching	After matching	0.673	0.792	−0.119	0.013	−8.89 ***
Radius matching	After matching	0.688	0.757	−0.069	0.016	−4.36 ***
K-nearest neighbour matching	After matching	0.692	0.748	−0.056	0.021	−2.62 ***
Kernel matching	After matching	0.688	0.758	−0.069	0.016	−4.37 ***

Note: *** represent significance at the levels of 1%, respectively.

**Table 6 ijerph-19-11955-t006:** Results of the mediation effect analysis.

Variant	Model (2)	Model (4)	Model (5)
	Energy Consumption Structure of Farmers	Gross Household Income	Energy Consumption Structure of Farmers
Nature reserve policy	−0.075 *** (−6.135)	0.222 *** (4.584)	−0.067 *** (−5.419)
Gross household income			−0.056 *** (−6.821)
Control variable	controlled	controlled	controlled
Constant	0.792 *** (15.465)	6.894 *** (19.959)	1.539 *** (14.902)
Sample size	956	956	956
R²	0.077	0.354	0.358
Adjusted R²	0.076	0.344	0.348

Note: (1) *** represent significance at the levels of 1%, respectively; (2) T values are mentioned in parentheses.

**Table 7 ijerph-19-11955-t007:** SUR test results.

Variant	Model (6)	Model (7)
Policy implications for nature reserves	−0.121 *** (−6.459)	−0.035 * (−1.953)
Control variable	Controlled	Controlled
*p*-value	0.002	
Number of samples	567	389
Constant	1.543 *** (10.833)	1.487 *** (9.861)
R²	0.390	0.350
Adjusted R²	0.373	0.323

Note: (1) *, *** represent significance at the levels of 10%, 1%, respectively; (2) T values are mentioned in parentheses; (3) ‘*p*-values’ were used to examine the significance of coefficient differences between the two groups; (4) controlled variables are the same as in Table 4, and only the regression results of the core variables are reported owing to space limitations.

## Data Availability

Not applicable.

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
