# Peer review of "Influence of Nature Reserves on the Energy Consumption Structure of Local Farmers"

_ijerph, 2022, doi:10.3390/ijerph191911955_

Round 1
Reviewer 1 Report
The article deals with the highly important topic of energy consumption in households. The article does not contain new, ground-breaking content. However, the article is well-written and easy to understand. Each stage of the research process has been adequately carried out and described.
However, there are some issues that could be improved:
1. No clearly defined aim of the study.
2. Page 9 unit of area (mu)? What is it?
3. In the introduction, it would have been useful to refer to studies conducted in other countries of the world if such exist.
4. Conlusion does not directly address the six research hypotheses set out.
Author Response
亲爱的编辑和审稿人,
非常感谢您抽出宝贵时间审阅此手稿。我们非常感谢您的所有慷慨意见和建议!请在下面找到我的逐项回复,并在重新提交的文件中找到我的修订。
问题1.该研究没有明确界定的目的。
回应:在引言的最后修改中,增加了本文的研究目的:本研究基于对辽宁省猴子岩、山大石、合商茅子省级自然保护区周围老图定子、白石、海唐山国家级自然保护区等农户的实地考察,采用OLS回归和PSM方法。探讨自然保护区政策对周边农户能源消费结构的影响,深入分析自然保护区的建立是否对农民消费结构有影响,验证农民环境认知的调节作用和农民家庭收入的中介作用对周边农户能源消费结构的影响机制。并提出政策建议,促进各地农民能源消费水平,倡导绿色能源消费。
问题2.第9页 面积单位(亩)?这是什么?
答:沐是中国市政制度下的一个土地面积单位,在中国农村常被用作面积单位。但根据审稿人的意见和学术论文的规格,本文以平方米为单位面积。通过单位换算(1亩=666.66m2),表3中的耕地面积和林地面积单位由之前的Mu(mu)改为平方米(m2)。
问题3.在导言中,如果存在的话,提及在世界其他国家进行的研究将是有益的。
响应:在介绍和讨论部分添加以下引用:
- 引用文章[11]家庭能源消耗与收入和相对生活水平:增加了小组方法以支持作者的观点。
- 引自文章 [14] 撒哈拉以南非洲城市和农村家庭能源转型:空间异质性是否揭示了转型方向?“家庭规模是决定农村家庭燃料选择的重要因素。
- 引用文章[18]农村家庭从传统能源向可再生能源过渡的偏好:北贡德尔地区能源阶梯假说的适用性“户主年龄增加1%,过渡性燃料使用增加0.04%。
- 引用文章[45] 安哥拉北部扎伊尔省的城市和农村家庭能源消耗和森林砍伐模式:景观方法“大多数贫困的农村居民通过长期使用生物质作为家庭能源来促进森林砍伐。
- 参见文章[50]非洲农村地区的可再生能源战略:以光伏为主导的可再生能源战略是向撒哈拉以南非洲农村贫困人口提供现代能源的正确方式吗?能量结构分类法。
- 引文[58] 《柬埔寨磅清省Des-Trict》中农村家庭能源消费结构的经济和环境成本 结论“在生物质作为主导能源结构下,每个家庭每月的现金支付较低,但环境成本最高。
问题4.结论并不直接解决提出的六个研究假设。
回应:本研究提出了4个假设。
假设1认为,自然保护区政策有利于降低传统能源消费在周边农户能源消费总量中的比重。对应结论:自然保护区对农民的能源消费结构有显著的负面影响,即自然保护区的监管政策有利于减少农民对传统生物质能的消费。这些政策通过行政手段孤立当地农民,禁止砍伐和使用山地森林的所有资源,并减少农民的传统能源消耗。
假设2认为,提高农民的环保意识将促进周边农民降低传统能源消费在能源消费总量中的比重。对应结论:环境认知作为调节变量,对降低农民传统生物质能消耗起到促进作用。环境认知作为农民减少低碳排放的重要心理动力,在减少高排放、高污染能源的消费、促进使用生态友好型能源等方面发挥着重要作用。
假说3认为,自然保护区通过增加家庭收入,间接降低了传统能源消费在周边农民总能耗中的比例。相应结论:农户总收入在自然保护区政策与农户能源消费结构的关系中起着中间作用。根据能源阶梯理论,随着收入的增加,农民从传统能源转向过渡能源,最终转向清洁能源,这既优于传统能源,也优于化石能源。
假设4:自然保护区水位越高,对周边农户能源消费结构的影响越明显。相应结论:自然保护区在影响农民传统生物质能消费方面存在一定的异质性。与居住在省级自然保护区周围的农民相比,居住在国家级自然保护区周围的农民对监管政策更为敏感。国家级自然保护区建立时间长,在一定程度上降低了农民对森林资源的高度依赖。此外,国家级自然保护区管理力度更强,相应的惩戒措施更严格,有效抑制了森林砍伐和森林盗窃。
再次感谢您和所有评论者的友好建议。
您真诚的,
李先生,
沈阳农业大学经济管理学院

Reviewer 2 Report
The idea of the article is good, as it aims to research the influence of human activity (in particular, farms) on the environment in a specific region and the change in climate in the whole world.
However, it is not clear from the text of the article which period is taken as the basis of the study (1981-2005 is indicated in Table 1), but already in the next section (3.2) the authors consider the data for 2020. It would be appropriate to take several time periods and analyze the dynamics of changes occurring in the studied periods.
In addition, the data are taken from only one country. It would be appropriate to consider similar regions of other countries and their methods and approaches to solving the problem formulated in the title of the article.
Only national literature is taken. There is only one reference from International sources of information (position in the list of literature - 51).
Also, it is not obvious from the title of the article which component of impact (economic, technical or social) the authors are considering (the authors are from the College of Economics and Management).
For data from tables, it would be worth taking bar charts for a visual representation and better visualization of information.
Author Response
亲爱的编辑和审稿人,
非常感谢您抽出宝贵时间审阅此手稿。我们非常感谢您的所有慷慨意见和建议!请在下面找到我的逐项回复,并在重新提交的文件中找到我的修订。
问题1.从文章的文本中看不出哪个时期是研究的基础(表1中指出了1981-2005年),但作者已经在下一节(3.2)中考虑了2020年的数据。应该花几个时间来分析所研究时期内发生的变化的动态。
回应:中国辽宁省有19个国家级自然保护区和27个省级自然保护区。本文随机选取了3个国家级自然保护区和3个省级自然保护区。表1显示了所调查的六个自然保护区的建立日期(1981-2005年)。本文使用的数据源是2021年6月至7月通过问卷进行的实地调查。本次调查选取了6个自然保护区6个县16个乡镇的44个村庄进行问卷调查,其中自然保护区内17个村,自然保护区外27个村。自然保护区内的村庄将受到自然保护区政策的影响,而自然保护区以外的村庄将不受自然保护区政策的影响。因此,本文根据“是否受自然保护区政策影响”将样本养殖户分为两组进行比较分析,得出了研究结论。
问题2. 此外,数据仅取自一个国家。应当考虑其他国家的类似区域及其解决该条标题中提出的问题的方法和途径。只拍摄国家文学。国际信息来源只有一个参考文献(在文献列表中的位置 - 51)。
响应:在介绍和讨论部分添加以下引用:
- 引用文章[11]家庭能源消耗与收入和相对生活水平:增加了小组方法以支持作者的观点。
- 引自文章 [14] 撒哈拉以南非洲城市和农村家庭能源转型:空间异质性是否揭示了转型方向?“家庭规模是决定农村家庭燃料选择的重要因素。
- 引用文章[18]农村家庭从传统能源向可再生能源过渡的偏好:北贡德尔地区能源阶梯假说的适用性“户主年龄增加1%,过渡性燃料使用增加0.04%。
- 引用文章[45] 安哥拉北部扎伊尔省的城市和农村家庭能源消耗和森林砍伐模式:景观方法“大多数贫困的农村居民通过长期使用生物质作为家庭能源来促进森林砍伐。
- 参见文章[50]非洲农村地区的可再生能源战略:以光伏为主导的可再生能源战略是向撒哈拉以南非洲农村贫困人口提供现代能源的正确方式吗?能量结构分类法。
- 引文[58] 《柬埔寨磅清省Des-Trict》中农村家庭能源消费结构的经济和环境成本 结论“在生物质作为主导能源结构下,每个家庭每月的现金支付较低,但环境成本最高。
问题3.此外,从文章的标题来看,作者正在考虑影响的哪个组成部分(经济,技术或社会)并不明显(作者来自经济与管理学院)。
应对:随着经济的快速增长,资源和能源的持续消耗、生态环境问题日益突出。为了解决这一重大问题,保护生态环境和自然资源,中国采取了一系列对策,其中建立自然保护区被视为保护生态系统的最佳方式。自然保护区提供的生物生态资源不仅是国家战略资源,也是人类生存和经济社会可持续发展的基础。随着自然保护区的建立,周围社区和农民不可避免地会产生经济、技术和社会影响。
经济影响:自然保护区建成后,为周边农民提供了更多的就业机会,改变了原有的生计模式,提高了他们的收入和消费水平。而且政府将限制农民的生产生活活动进行相应的经济补偿。
技术影响:自然保护区的建立促进了当地农业技术创新。
社会影响:自然保护区的建立改善了周边社区的资源环境,增强了自然保护区周边农民的幸福感。
本文主要考虑了作为自然保护区产生的经济影响一部分的能源消费结构,揭示了自然保护区对周围农户能源消耗的影响及其影响机制。
您的评论对我未来的研究非常有帮助,我将继续在后续研究中结合技术影响和社会影响进行深入研究。
问题4.对于表格中的数据,值得采用条形图来直观表示和更好地可视化信息。
回应:为了将描述性统计和分析结果的信息可视化,根据审稿人的意见,将表4替换为直方图图3。
再次感谢您和所有评论者的友好建议。
您真诚的,
李先生,
沈阳农业大学经济管理学院

Round 2
Reviewer 2 Report
It is very good that the authors have studied many sources on the topic of the study, however, further on, it is recommended to limit ourselves to the main ones (currently, references to sources make up 3 pages of the publication).
The use of bar charts significantly improved the perception of analytical data. It would be worth using them in the visualization of further studies.
Also, in future studies, it would be worthwhile to compare the results obtained in the course of this study with world practices in this field.
Author Response
Dear Editors and Reviewers,
Thanks very much for taking your time to review this manuscript. We really appreciate all your generous comments and suggestions! Please find my itemized responses in below and my revisions in the re-submitted files.
Question 1. It is very good that the authors have studied many sources on the topic of the study, however, further on, it is recommended to limit ourselves to the main ones (currently, references to sources make up 3 pages of the publication).
Response:According to the comments of the reviewers, the authors cut 19 references and kept 39 main citation sources.
Question 2. The use of bar charts significantly improved the perception of analytical data. It would be worth using them in the visualization of further studies.
Response:Thank you very much for your suggestions on this paper. After modification according to your requirements, compared with the table, the difference of energy consumption in the nature reserve and outside the nature reserve can be found more intuitively through bar Figure 3. In this paper, there is no table that can be further visualized, so the author does not modify other tables.
Question 3. Also, in future studies, it would be worthwhile to compare the results obtained in the course of this study with world practices in this field.
Response:Thank you very much for your suggestion, the author added this flaw to the research deficiency, and the results will be compared with world practice in this field in future research.
Thank you and all the reviewers for the kind advice again.
Yours sincerely,
Mr. Li,
College of Economics and Management, Shenyang Agricultural University
